# TransformEHR: transformer-based encoder-decoder generative model to enhance prediction of disease outcomes using electronic health records

Zhichao Yang [1], Avijit Mitra[1], Weisong Liu[2,3], Dan Berlowitz[3,4] &
Hong Yu [1,2,3,5] ✉

Deep learning transformer-based models using longitudinal electronic health records (EHRs) have shown a great success in prediction of clinical diseases or outcomes. Pretraining on a large dataset can help such models map the input space better and boost their performance on relevant tasks through finetuning with limited data. In this study, we present TransformEHR, a generative encoder-decoder model with transformer that is pretrained using a new pre-training objective—predicting all diseases and outcomes of a patient at a future visit from previous visits. TransformEHR's encoder-decoder framework, paired with the novel pretraining objective, helps it achieve the new state-of-the-art performance on multiple clinical prediction tasks. Comparing with the previous model, TransformEHR improves area under the precision–recall curve by 2% ($p < 0.001$) for pancreatic cancer onset and by 24% ($p = 0.007$) for intentional self-harm in patients with post-traumatic stress disorder. The high performance in predicting intentional self-harm shows the potential of TransformEHR in building effective clinical intervention systems. TransformEHR is also generalizable and can be easily finetuned for clinical prediction tasks with limited data.

The widespread adoption of electronic health records (EHRs) among the US hospitals has led to the development and adoption of numerous data mining and statistical techniques for EHRs. Longitudinal EHRs have been successfully used to predict clinical diseases or outcomes[1-4]. Early work applied regression and traditional machine learning (ML) based models (e.g., support vectors machines, random forest, and gradient boosting) to predict single disease or outcome. Examples include congestive heart failure[5], sepsis mortality[6], mechanical ventilation[6], septic shock[7], type 2 diabetes[8], and development of post-traumatic stress disorder (PTSD)[9], among others.

With the availability of large cohorts and computational resources, deep learning based models can outperform traditional ML models[10-16]. State-of-the-art (SOTA) models in EHR-based predictive modeling achieved this through the pretrain-finetune paradigm - a two-step process where the model is first trained on large-scale longitudinal EHRs to learn the representations of clinical features such as, International Classification of Diseases (ICD) codes (pretrain) and then further trained to adapt to specific tasks e.g., outcome prediction (finetune). Models such as Med-BERT[13], BEHRT[14], and BRLTM[15] fall in this category. However, their pretraining objectives were limited in

[1]College of Information and Computer Science, University of Massachusetts Amherst, Amherst, MA, USA. [2]School of Computer & Information Sciences, University of Massachusetts Lowell, Lowell, MA, USA. [3]Center for Healthcare Organization and Implementation Research, VA Bedford Health Care System, Bedford, MA, USA. [4]Department of Public Health, University of Massachusetts Lowell, Lowell, MA, USA. [5]Center for Biomedical and Health Research in Data Sciences, University of Massachusetts Lowell, Lowell, MA, USA. ✉e-mail: Hong_Yu@uml.edu

predicting a fraction of ICD codes within each visit. In reality, most patients have multiple diseases or outcomes at once[17], many of which are highly correlated (such as obesity, diabetes, and hypertension[18–20]) and thus collectively contribute to the disease or outcome trajectories. Therefore, a novel pretraining strategy, which predicts the complete set of diseases and outcomes within a visit, might improve clinical predictive modeling.

In this study, we propose TransformEHR, an innovative denoising sequence to sequence transformer[21] model that was pretrained on 6.5 million patients' EHRs to predict complete ICD codes of a visit. TransformEHR can be further finetuned for single disease or outcome predictions. Unlike previous EHR-based models[13–16] which rely on the bidirectional (left-to-right and right-to-left) encoder representation from transformers (BERT) framework[22], TransformEHR used a transformer-based encoder-decoder generative framework to predict future ICD codes during pretraining. The unidirectional (left-to-right) decoder in such an encoder-decoder framework is more similar to the use case of future disease or outcome predictions based on history of past diseases or outcomes (past-to-future) compared to the bidirectional encoder-only framework.

Although the encoder-decoder framework was originally designed to generate next sentence given previous sentences as context[23,24], we repurposed the framework for TransformEHR to generate the ICD codes of the next visit given previous EHRs (context). TransformEHR can utilize cross-attention[21] by identifying relevant ICD codes from previous visits to predict future ICD codes. The decoder then predicts ICD codes one after another by using already predicted diagnostic ICD codes to predict next ICD codes. Furthermore, TransformEHR includes date of each visit to integrate temporal information, whereas previous transformer-based predictive models only included their sequential order[13–16]. Specific date of each visit is an important feature in predictive modeling as importance of predictor in a visit can vary over time[1,25–27].

We evaluated TransformEHR for a broad range of disease and outcome predictions. In addition to predictions of ICD codes, we evaluated TransformEHR on two challenging and clinically important disease and outcome prediction tasks: pancreatic cancer prediction and intentional self-harm prediction among PTSD patients. In summary, our key contributions are as follows:

First, we propose a new pretraining objective that predicts all diseases or outcomes of a future visit using longitudinal information from the previous visits. Such a pretraining objective helps TransformEHR uncover the complex interrelations among different diseases and outcomes.

Second, this is the first study that explored a generative encoder-decoder framework to predict patients' ICD codes using their longitudinal EHRs. Our encoder-decoder framework outperformed the encoder-based models in part due to the decoder self-attention and cross-attention mechanisms. TransformEHR outperformed SOTA BERT models on both common and uncommon ICD code predictions. In particular, the improvements for uncommon ICD code predictions were substantial.

Third, TransformEHR achieved a positive predictive value (PPV) of 8.8% for prediction of intentional self-harm among the top 10% PTSD patients at high predicted risk. A recent study has shown that a practical suicide prevention tool must achieve above 1.7% PPV to be considered as cost-effective: balance the costs of providing the intervention against the potential health care related costs if self-harm occurs[28]. A PPV of 8.8% is substantially above the threshold of 1.7% needed to balance the cost in clinical practice. This shows the potential of deploying TransformEHR for clinical screening and interventions.

Finally, we validated the generalizability of TransformEHR using both internal and external datasets. Internally, we evaluated TransformEHR on never-seen in-domain EHR data from Veterans Health Administrations (VHA) facilities. Externally, we evaluated TransformEHR on out-of-domain data from a non-VHA hospital. Our results demonstrated a strong transfer learning[29] capability of TransformEHR, which can greatly benefit hospitals with limited data and computing resources.

## Results
### Data
Our pretraining cohort comprises 6,475,218 patients who received care from more than 1200 health care facilities of the US VHA from 1/1/2016 to 12/31/2019. To evaluate pretrained models, we created two disease/outcome agnostic prediction (DOAP) datasets—one for common and one for uncommon diseases/outcomes. We selected 10 ICD-10CM codes with the highest prevalence (prevalence ratio >2%) in our pretraining cohort for our common disease/outcome DOAP dataset. As for the set of uncommon diseases/outcomes, we followed the FDA guidelines[30] to randomly select 10 ICD-10CM codes with a prevalence ratio ranging from 0.04% to 0.05% in our pretraining cohort. The lists of common and uncommon diseases/outcomes are shown in Table 1. These codes were assigned by VHA medical record technician and served several important purposes including clinical studies, performance measurement, workload capture and operation, cost determination, and billing. To assess the generalizability, we evaluated TransformEHR on out-of-domain non-VHA EHR data. We used the MIMIC-IV dataset[31] to build a non-VHA DOAP dataset. The MIMIC-IV dataset includes intensive care unit patients admitted to the Beth Israel Deaconess Medical Center in Boston, Massachusetts. Since the dataset contains information from 2008 to 2019 but the implementation of ICD-10CM started from October 2015, we only selected patients with the ICD-10CM records to match our implementation for the cohorts from VHA, resulting in a dataset of 29,482 patients.

To evaluate finetuned models for single disease and outcome predictions with low prevalence ratio, we created two EHR datasets for two important prediction tasks: pancreatic cancer (single disease) and intentional self-harm among patients with PTSD (single outcome). Early screening, early diagnosis and early treatment of pancreatic

**Table 1 | Disease or outcome definitions in this study**

| Task | Disease or Outcome (ICD-10CM code) |
|---|---|
| Prediction of single disease or outcome | New onset pancreatic cancer (C25) Intentional self-harm among patients with PTSD |
| Disease/outcome agnostic prediction - common | Chronic post-traumatic stress disorder (F43.12) Type 2 diabetes (E11.9) Hyperlipidemia (E78.5) Loin pain (R10.3) Low back pain (M54.50) Obstructive sleep apnea (G47.33) Depression (F33.9) Obstructive airway disease (J44.9) Gastroesophageal reflux disease (K21.9) Arteriosclerosis (I25.10) |
| Disease/outcome agnostic prediction - uncommon | Benign neoplasm of connective tissue of eyelid (D21.0) Refractory anemia (D46.4) Melanocytic nevi of upper limb (D22.6) Benign neoplasm of skin of upper eyelid (D23.10) Cutaneous abscess of axilla (L02.41) Ankle and foot subacute osteomyelitis (M86.27) Cortical hemisphere nontraumatic hemorrhage intracerebral (I61.1) Malignant neoplasm of head of pancreas (C25.0) Other complication of kidney transplant (T86.19) Nonexudative age-related macular degeneration (H35.31) |

cancer are critical for this deadly disease[32, 33]. Med-BERT[13] has been used to predict pancreatic cancer. Intentional self-harm is common among the US military Veterans with PTSD[34]. The detailed pretraining and finetuning cohort definitions are presented in Methods section. We compare these cohorts in Supplementary Table 1. To assess the generalizability, we also evaluated TransformEHR on Veterans with PTSD from VHA facilities not included in the pretraining cohort.

## Longitudinal EHRs

As shown in Fig. 1, TransformEHR takes longitudinal EHRs as input. To compare TransformEHR with the previous SOTA EHR-based models using BERT[13,15], we included demographic information and ICD-10CM codes as predictors. Demographic information includes gender, age, race, and marital status. The attributes (e.g., male) of each category (e.g., gender) were appended as individual predictors. Following previous work[13], we grouped ICD codes at visit level. Within a visit, we ordered ICD codes from high to low priority, as assigned by health care providers, where the primary diagnosis is typically given the highest priority, followed by the secondary diagnosis and so on.

Multiple visits of each patient formed a time-stamped (by date of visit) input of a sequence of ICD-10CM code groups (Fig. 2a). We used multi-level embeddings[13]. Embeddings are trainable fixed-dimensional vectors that were used to represent predictors in a continuous vector space and were learned during the pretraining process (details in the next section). We represented each visit in an input sequence using a visit embedding and each ICD code using a code embedding. To embed the time, we applied sinusoidal position embedding[21] to the numerical format of visit date (date-specific). However, the use of actual visit dates, which is protected health information sensitive, may impact the deployability of the model. Thus, we also explored using the relative time information −days difference to embed time (days-diff). Specifically, we calculate the days difference between a certain visit and the last visit in the EHR. Finally, each input embedding was constructed by summing up the corresponding code embedding, visit embedding, and time embedding (Fig. 2a).

## Pretrain-Finetune paradigm

With longitudinal EHRs as input, we first pretrained models on the pretraining cohort of 6,475,218 patients and then finetuned the model for single disease or outcome prediction, as shown in Fig. 3. During the pretraining, the model was trained to recover the original longitudinal EHRs from corrupted (masked) longitudinal EHRs.

Previous EHR-based BERT models corrupted longitudinal EHRs by randomly sampling 25% ICD codes and replacing them with mask (code masking)[13–15]. TransformEHR, on the other hand, masked all ICD codes in a single visit (visit masking). A comparative example is illustrated in Fig. 2b. In other words, TransformEHR was pretrained to predict the complete set of ICD codes of a patient's future visit, given demographic information and longitudinal ICD codes up to the current visit.

## TransformEHR architecture

TransformEHR uses an encoder–decoder transformer architecture[23,24]. The encoder processes the input embeddings and generates a set of hidden representations for each predictor. Unlike the encoder-only transformer architecture used by existing EHR-based BERT models[13–15], TransformEHR performs cross-attention over the hidden representations from the encoder and assigns an attention weight for each representation. These weighted representations are then fed to the decoder, which generates ICD codes of the future visit as illustrated in Fig. 3. The decoder generates ICD codes following the sequential order of code priority within a visit. In other words, it first generates primary diagnosis, and then generates secondary diagnosis based on primary diagnosis. This step is repeated until all diagnoses of a future visit are

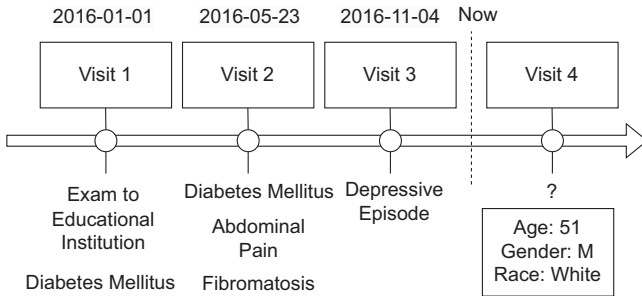

**Fig. 1 | Exemplar EHR sequence of a patient (white, male, age 51).** This EHR contains demographic and ICD codes from 3 previous visits, including Z-codes such as exam to educational institution Z02.0. The pretraining objective is to predict all diseases and outcomes in the next visit. All these values are artificial and for illustration purposes.

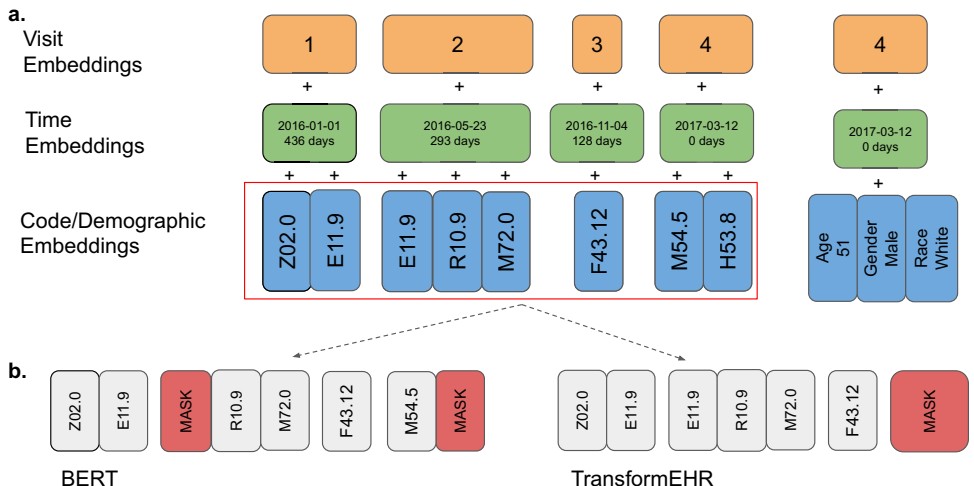

**Fig. 2 | How model learns the correlation of ICD codes by recovering the masked ICD codes to its original ICD codes. a** illustrates the preprocessing of input data from 4 patient visits, with visit embeddings capturing chronological information, time embeddings capturing the specific date, and ICD codes along with demographic data encoded as embeddings. **b** displays various medical history masking schemes: BERT-style masking random codes and TransformEHR-style masking the all codes in a visit.

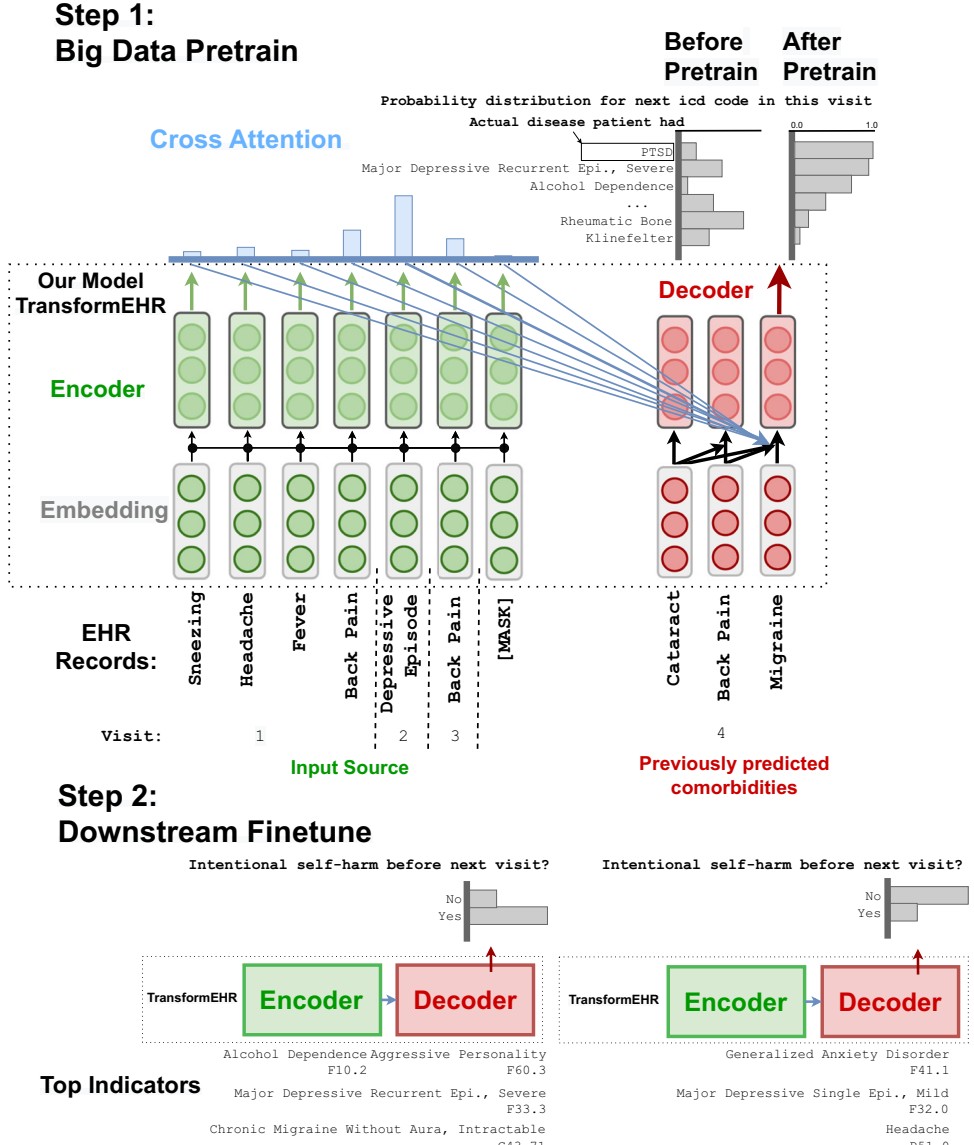

**Fig. 3 | TransformEHR architecture and pretrain-finetune paradigm.** During *Step 1*, TransformEHR was pretrained with generative encoder-decoder transformer on a large set of longitudinal EHR data. TransformEHR learned the probability distribution of ICD codes (vs. random distribution) through correlation of cross attention. During *Step 2*, we then finetuned TransformEHR to the predictions of single disease or outcome. Through attention weights, TransformEHR was able to identify top indicators for the predictions. Encoder is colored in green, decoder is colored in red, and cross attention that connects both is colored in blue.

completed. Our results showed that the TransformEHR architecture outperformed the encoder-only architecture.

### Evaluation metrics

We reported PPV (precision), area under the receiver operating characteristic curve (AUROC), and area under the precision recall curve (AUPRC) to measure models' performance on single disease or outcome predictions. PPV is the fraction of true positives from all predicted positives. AUROC is the area under sensitivity and false positive rate curve. Sensitivity (or recall) is the number of true positives divided by the number of ground truth positives. False positive rate is the number of false positives divided by total number of negatives. AUPRC is the area under PPV and sensitivity curve, and it has shown to be an effective measure for highly imbalanced binary classification problems, which include self-harm prediction[35,36]. We compared TransformEHR with four strong baseline models: logistic regression, long short-term memory (LSTM)[37], BERT[22] without pretraining, and BERT[22] pretrained on VHA cohort.

### Pretraining evaluation: disease or outcome agnostic prediction

DOAP is the task of predicting the ICD codes of a patient's future visit based on patient's demographic information and longitudinal ICD codes up to the current visit. Compared to BERT, TransformEHR improved the AUROC in all prediction subtasks, regardless of disease/outcome category (common or uncommon) and occurrence type (new or recurrent), as shown in Table 2. We found a 3.96% relative increase in common diseases/outcomes and 5.92% in uncommon diseases/outcomes.

Our TransformEHR contains three unique components compared to previous medical BERT-based models: (1) visit masking, (2) encoder-decoder architecture, and (3) time embedding. We performed ablation analysis to evaluate the effectiveness of each component. First, we compared an encoder-decoder model that uses visit masking, i.e., masking all ICD codes of a visit, to another encoder-decoder model that uses code masking, i.e., masking a randomly selected fragment of ICD codes of a visit. Our findings revealed that the visit masking performed better (showing an improvement of 95%CI: 2.52%–2.96%,

**Table 2 | Disease/outcome agnostic prediction: AUROC scores on different pretraining objectives for the 10 common and 10 uncommon diseases in Table 1**

| Models | | BERT | | TransformEHR | |
|---|---|---|---|---|---|
| Chronic PTSD | R | 81.04 | ±0.11 | 83.73 | ±0.07 |
| | O | 76.74 | ±0.17 | 77.95 | ±0.12 |
| Type 2 diabetes | R | 85.00 | ±0.10 | 85.72 | ±0.07 |
| | O | 79.97 | ±0.04 | 81.84 | ±0.05 |
| Hyperlipidemia | R | 86.78 | ±0.03 | 88.04 | ±0.05 |
| | O | 81.28 | ±0.08 | 83.42 | ±0.08 |
| Loin pain | R | 81.47 | ±0.04 | 88.24 | ±0.05 |
| | O | 76.88 | ±0.12 | 85.37 | ±0.08 |
| Low back pain | R | 85.43 | ±0.07 | 86.94 | ±0.03 |
| | O | 80.16 | ±0.07 | 82.30 | ±0.10 |
| Obstructive sleep apnea | R | 80.74 | ±0.17 | 82.25 | ±0.16 |
| | O | 73.06 | ±0.08 | 74.69 | ±0.19 |
| Depression | R | 86.73 | ±0.05 | 87.66 | ±0.12 |
| | O | 82.60 | ±0.12 | 83.85 | ±0.11 |
| Obstructive airway disease | R | 83.57 | ±0.14 | 86.19 | ±0.07 |
| | O | 76.99 | ±0.08 | 80.27 | ±0.07 |
| Gastroesophageal reflux | R | 84.98 | ±0.28 | 91.07 | ±0.11 |
| | O | 76.29 | ±0.36 | 83.41 | ±0.33 |
| Arteriosclerosis | R | 82.21 | ±0.06 | 88.79 | ±0.10 |
| | O | 75.78 | ±0.08 | 80.03 | ±0.20 |
| Uncommon disease/outcome | O | 75.63 | ±0.12 | 80.11 | ±0.12 |

Many common diseases are chronic in nature. We therefore study whether prior history of the same disease has an impact on prediction performance, where R is recurrent and O is new disease onset. ± represents standard deviation.

**Table 3 | Performance of models for pancreatic cancer prediction**

| Models | | AUROC | | AUPRC | |
|---|---|---|---|---|---|
| Without Pretraining | Logistic regression | 73.64 | ±2.26 | 68.95 | ±2.14 |
| | LSTM | 76.98 | ±0.54 | 73.48 | ±0.55 |
| | BERT without pretraining | 77.27 | ±0.45 | 74.00 | ±0.31 |
| With Pretraining | BERT | 79.22 | ±0.47 | 76.89 | ±0.48 |
| | TransformEHR (ours) | 81.95 | ±0.90 | 78.64 | ±0.85 |

Result is calculated from best hyperparameters with 5 randomized seeds each. ± represents standard deviation.

**Table 4 | Performance (and standard deviation) of predictive models for intentional self-harm**

| Models | | Self-Harm w/ Short History | | Self-Harm w/ Full History | |
|---|---|---|---|---|---|
| | | AUPRC | AUROC | AUPRC | AUROC |
| Without Pretraining | Logistic regression | 6.89 | 66.87 | 3.15 | 64.60 |
| | | ±1.55 | ±0.60 | ±0.77 | ±3.73 |
| | LSTM | 9.13 | 71.46 | 8.36 | 69.36 |
| | | ±0.74 | ±0.13 | ±0.80 | ±0.83 |
| | BERT without pretraining | 9.39 | 71.78 | 10.98 | 72.53 |
| | | ±0.30 | ±0.18 | ±0.66 | ±0.69 |
| With Pretraining | BERT | 10.30 | 71.87 | 13.34 | 78.02 |
| | | ±0.83 | ±0.79 | ±1.34 | ±1.84 |
| | TransformerEHR | 13.77 | 74.89 | 16.67 | 79.90 |
| | | ±0.69 | ±0.77 | ±1.56 | ±1.73 |

"Self-Harm w/ Full History" refers to cases where the prediction is based on the original EHR (mean: 10.1 visits, st.dev.: 3.3 visits) prior to predicting intentional self-harm. "Self-Harm w/ Short History" includes only the 5 most recent visits. ± represents standard deviation. Result is calculated from best hyperparameters with 5 randomized seeds each.

$p < 0.001$ in AUROC) compared to code masking for all diseases/outcomes tested as shown in Supplementary Table 2. These results demonstrated that pretraining for prediction of all diseases and outcomes outperform traditional pretraining objectives.

Next, we compared encoder-decoder model with encoder-only model (BERT) on DOAP. Our findings showed that the encoder-decoder model outperformed the encoder-only model (with an improvement of 95%CI: 0.74%–1.16%, $p < 0.001$ in AUROC) across all diseases/outcomes tested (Supplementary Table 2).

Finally, we conducted an experiment by excluding time embedding from the implementation of TransformEHR. Time embedding allows TransformEHR to capture the temporal information of prior visits. Our results indicated that TransformEHR with date-specific time embedding exhibits a moderate improvement (with an increase of 95% CI: 0.01%, 0.43%, $p = 0.021$ in AUROC) than TransformEHR without time embedding across most diseases/outcomes (Supplementary Table 2). Instead of specific date as time embedding, we also tried days difference between visits as time embedding, TransformEHR with date-specific embedding showed no significant improvement than TransformEHR with days difference embedding. Thus, we chose days difference embedding for later experiments as it provides better protection to patient health information compared to the specific date.

**Finetuning evaluation: single disease or outcome prediction**

Results of pancreatic cancer onset prediction are shown in Table 3. On the metric of AUROC, TransformEHR achieved 81.95 (95% CI: 81.06, 82.85) outperforming both the logistic regression model (73.64; 95% CI: 71.39, 75.90, $p < 0.001$), LSTM (76.98; 95% CI: 76.44, 77.52, $p < 0.001$) and BERT (79.22; 95% CI: 78.75, 79.69, $p < 0.001$). A similar trend is observed on AUPRC. TransformEHR achieved AUPRC of 78.64

(95% CI: 77.80, 79.49) outperforming both the logistic regression model (68.95; 95% CI: 66.81, 71.08, $p < 0.001$), LSTM (73.48; 95% CI: 72.93, 74.03, $p < 0.001$) and BERT (76.89; 95% CI: 76.41, 77.37, $p < 0.001$).

Predicting intentional self-harm in patients with PTSD is challenging because of its low prevalence (1.9%). As shown in Table 4, AUPRC of TransformEHR was 16.67 (95% CI: 15.11, 18.23, $p = 0.007$), outperforming BERT (13.34; 95% CI: 12.00, 15.11, $p < 0.001$), LSTM (8.36; 95% CI: 7.56, 9.16, $p < 0.001$), and logistic regression (3.15; 95% CI: 2.39, 3.92, $p < 0.001$) by 24%, 97%, and 422% respectively. A similar trend is observed for AUROC. We also calculated sensitivity and PPV for a variety of thresholds. As shown in Supplementary Table 3, The PPV of TransformEHR ranges from 3.14 to 8.80 for 10% to 60% threshold.

**Subgroup analyses among different demographics.** The results of the subgroup analyses are shown in Supplementary Table 4. AUPRC was consistent among different genders, ages, races, and marital status. For example, the AUPRC of patients who were more than 80-year-old (4.86% of the cohort) was 18.08 (95% CI, 16.05, 19.66), which was not statistically different from the AUPRC of patients whose ages range from 30 to 39 (19.58% of the cohort): 16.29 (95% CI, 14.31, 18.27).

**Effect of historical EHR length on performance.** To examine the impact of how patients' prior history impacts model predictions, we conducted two experiments - "self-harm with short history" (included only the five most recent visits prior to self-harm) and "self-harm" included all visits prior to self-harm. As shown in Table 4, The AUPRC of TransformEHR using only the five prior visits was 13.77. Using all visits, the AUPRC improved by 19% to 16.67.

## Generalizability evaluation

We validated the generalizability of TransformEHR both internally and externally. Out of 1239 VHA health care facilities, we found that EHR data from 121 facilities were not included in the pretraining cohort. We created an internal generalizability evaluation dataset using this subset of EHR data for the intentional self-harm prediction task among PTSD patients. Upon evaluating TransformEHR on this dataset, our results showed no statistical difference in AUPRC between data from unseen facilities (1st bar on the left of Supplementary Fig. 1) and data from facilities where at least some of the data were included in the pretraining cohort (other bars of Supplementary Fig. 1).

To evaluate TransformEHR's generalizability on the external dataset, we further finetuned the pretrained TransformEHR and BERT on the training set of a non-VHA DOAP dataset and evaluated them on the testing set. The distribution of the ICD codes assigned to an ICU population would be different from the ICD distribution in the VA, as shown in Supplementary Fig. 2. The evaluation result is shown in Supplementary Table 5. Compared to TransformEHR without pretraining, TransformEHR with pretraining improved AUROC by 2.3% (95% CI, 0.8%, 3.6%, $p = 0.005$). In comparison, BERT with pretraining outperformed BERT without pretraining in AUROC by 1.2% (95% CI, −0.3%, 2.7%, $p = 0.103$). The comparison of performance gain shows that TransformEHR offers better generalizability on the external dataset compared to BERT.

## Discussion

In this study we introduced TransformEHR, a generative deep neural network model for the prediction of diseases and outcomes using patients' longitudinal EHRs. By first pretraining TransformEHR on a large collection of EHRs (255 million visits from 6.5 million patients between 2016 and 2019) and then finetuning for specific clinical applications, we found that TransformEHR outperformed existing SOTA models on a wide range of disease or outcome predictions. The performance gain was substantial on intentional self-harm prediction among PTSD patients. As shown in Table 4, TransformEHR was the best-performing model and outperformed BERT, LSTM, and logistic regression models by 24%, 97%, and 422% respectively. The aforementioned results were not surprising, as deep-learning-based models are known to capture the salient information of EHR data to create powerful feature representations[38]. In addition, pretraining on large data and using the encoder-decoder framework have both been shown to be SOTA strategies for sequence-to-sequence applications[23].

TransformEHR outperformed BERT for prediction of both single and multiple diseases/outcomes, as shown in Tables 2, 3, and 4. In contrast to BERT, TransformEHR learned the representation of each clinical variable by predicting the complete diseases and outcomes with the help of cross-attention and decoder self-attention. Cross-attention allowed TransformEHR to selectively focus on sections of past visit ICD codes that were most relevant to predicting each ICD code in the future visit, and the decoder self-attention helped TransformEHR in predicting complete diseases and outcomes by leveraging already predicted primary diagnostic ICD codes to predict secondary codes that are less common. Thus, the performance gains were the highest among uncommon disease predictions. As shown in Table 2, compared with the BERT model, TransformEHR improved the AUROC by an average of 3.96% in predicting 10 common diseases, and by an average of 5.92% in predicting 10 uncommon diseases. TransformEHR also substantially improved prediction for intentional self-harm prediction (AUPRC from 13.34 to 16.67, $p = 0.007$) and pancreatic cancer prediction (AUPRC from 76.89 to 78.64, $p < 0.001$).

Pretraining played a key role in improving the performance of our deep-learning-based models. To demonstrate, we chose intentional self-harm prediction and compared the performance between a BERT model pretrained on our pretraining cohort (as described in the Data section) and a BERT model with no pretraining (parameters were randomly initialized before fine-tuning). Pretrained BERT substantially outperformed the non-pretrained BERT (23% higher AUPRC and 7% higher AUROC), as shown in Table 4. In particular, pretraining helps improve the latent representations of EHR data compared to non-pretrained models. This helped improve the probability distribution of candidate diseases or outcomes. As shown in Fig. 3, after pretraining on a large EHR cohort, the probability distribution for the next visit ICD code changed from a random distribution to a learned representation of clinically relevant diseases or outcomes. While a pretrained model can capture the probability distribution at a large cohort level, fine-tuning it further can improve the performance for a specific application.

Attention-based models benefit from longitudinal EHRs with long histories (approximately 10 prior visits). As shown in Table 4, of the attention-based models (TransformEHR and BERT) finetuned on more than 5 visits, the AUPRC scores improved by 19% and 31%, respectively, in comparison with models finetuned on only five most recent visits. However, with the same experimental setup, the AUPRC scores of non-attention-based models such as LSTM and logistic regression decreased by 8% and 54%, respectively, when finetuned on more than five prior visits (on average 10) compared with the models finetuned on only the five most recent visits. These findings were consistent with previous research[39], which showed that LSTM, although good at mitigating the vanishing gradient challenge of RNN, remains suboptimal with long-time dependencies.

Our work is also related to predictive model studies focused on intentional self-harm[40–44]. Typically, these approaches sample thousands of patients and use probability tables, decision trees, and logistic regression to predict intentional self-harm. PPV plays a crucial role here in estimating the potential benefit of any intensive case management intervention, if one were to be implemented based on any such approach. PPV indicates the percentage of patients receiving an intervention who would otherwise attempt self-harm. Hartl et al. [42]. predicted intentional self-harm among PTSD patients (prevalence ratio 5%, PPV 0%). Simon et al. [44]. integrated EHR data and questionnaires for 2,960,929 patients to predict suicide attempts (prevalence ratio 1%) within 90 days of a mental health visit. The most successful model demonstrated a PPV of 5%. TransformEHR, on the other hand, achieved a PPV of 8.80% at 10% threshold for prediction of intentional self-harm among PTSD patients, which substantially outperformed the baseline (Supplementary Table 3). In other words, out of the 100 highest-predicted-risk patients from 1000 previously diagnosed PTSD patients, 9 patients would attempt intentional self-harm for the first time before the next visit. In comparison, for logistic regression the PPV at the 10% threshold was 3.31%. A practical suicide prevention tool must have a relatively high PPV to minimize the resource cost and intrusion directed at patients who will never attempt self-harm[45]. Following the previous work[28], we calculated the incremental cost-effectiveness ratio (ICER) of self-harm risk prediction and intervention: the ratio of its incremental cost to its incremental quality-adjusted life-years (QALYs) compared with usual care. Previous studies in the US have suggested that ICER thresholds under $150 k per QALY can be used to determine the cost-effectiveness of healthcare interventions in 2014[46]. When using cognitive behavioral therapy as intervention, BERT achieves ICER of $123 k per QALY, and TransformEHR could reduce ICER to $109 k per QALY. Therefore, TransformEHR may be a reliable and feasible option for designing an effective suicide prevention system.

In conclusion, our results have multiple clinical implications. First, existing predictive models tend to focus on single diseases. Yet the focus in clinical care, particularly in older people, is often on managing comorbidities[17]. A predictive model that can predict multiple diseases or outcomes may be useful in designing complex treatment plans. Second, a disease-specific approach could require building hundreds of different predictive models, an inefficient and costly use of

resources whereas TransformEHR's unique pretraining objective enables it to predict all diseases and outcomes of a visit, immediately after the pretraining. Third, TransformEHR outperformed existing SOTA predictive models with a substantial performance gain, especially for uncommon disease/outcome predictions. Thus, our approach could assist in the development of screening algorithms to detect uncommon conditions. Fourth, TransformEHR can be easily generalized on an out-of-domain dataset with a significantly smaller training data. TransformEHR's strong transfer learning capability makes it a good fit for hospital settings with limited data and computing resources.

Despite the merits of this study, there are several limitations that provide opportunities for future improvements. First, we followed previous work on model pretraining and included only diagnostic ICD codes and demographic information[13]. Other information such as procedure codes, medications, lab results, and phenotypical information extracted from notes can be added to further improve the performance[47]. Combining all these codes together would result in a large vocabulary, forcing the model to have a huge embedding matrix as the input and output layer. The computation on this matrix would cause both a significant increase of memory and time complexity. In future work, we will increase the GPUs resource to mitigate this computational challenge. Second, our prediction of single disease was limited to pancreatic cancer and single outcome was limited to intentional self-harm with PTSD. Future works would expand the set of diseases and outcomes. Third, while we followed the previous studies to use ICD-10-CM codes to identify intentional self-harm[44,48,49], we acknowledge that the ICD-10-CM representation of self-harm would miss some, although a small percentage of patients who conducted self-harm[50]. In addition, the date of the encounter may not be the actual date of the self-harm. To determine the viability of using TransformEHR in a pragmatic trial, a tailored cost-effectiveness analysis that addresses these issues would be necessary for future work.

## Methods

### VHA cohort
Using the VHA Corporate Data Warehouse (CDW), we first identified a total of 8,308,742 patients who received care from more than 1200 health care facilities of the US VHA from 1/1/2016 to 12/31/2019. For inclusion, we required each patient to have at least two visits: one outcome visit and at least one prior visit to be used for prediction of the outcome. This resulted in a total of 6,829,064 patients.

### Pretraining and finetuning cohorts
Following the standard 95–5 split[13,22], we randomly took 95% of the patients (6,475,218) to create our pretraining cohort and used the remaining 5% (353,846) to create other datasets to finetune for DOAP and single disease or outcome predictions. The detailed patient cohort selections and diseases or outcomes for this study are shown in Supplementary Fig. 3. The study protocol was approved by the Institutional Review Board at the VA Bedford Healthcare System.

We identified two use cases to evaluate TransformEHR.

### Pancreatic cancer
Although pancreatic cancer is relatively uncommon, it is deadly, projecting to become the second leading cause of cancer-related mortality by 2030[32]. Since effective screening is not available for pancreatic cancer, most patients can seek medical attention only after being diagnosed with locally advanced or metastatic cancer. Therefore, accurate prediction of pancreatic cancer can help early detection of pancreatic cancer. Previous research[13] has evaluated BERT for pancreatic cancer prediction. Following the same criteria in previous studies[13,51], we defined pancreatic cancer with the first 3 digit ICD-10CM codes as C25. The case included 4639 patients of 45 years or older with no report of any other cancer disease before their first pancreatic

cancer diagnosis and diagnosis made between 12 and 36 months after their last visit, and control patients comprised 5089 patients of 45 years or older without any cancer diagnosis.

### Prediction of intentional self-harm
PTSD is considered as a hallmark injury of US Post-9/11 Veterans, with a prevalence estimated to be up to 23%[52]. Individuals with PTSD also have co-occurring physical health (e.g., chronic pain[53]), mental health (e.g., depression and anxiety[54,55]), and behavioral conditions (e.g., substance use disorder[56,57]). People with PTSD have 5.3–13 times the rate of suicide than people without PTSD[58]. Hence, we also evaluated TransformEHR for prediction of intentional self-harm for patients with PTSD. Out of 70,967 the patients who had been diagnosed with PTSD from 353,846 patients not in the pretraining cohort, 1342 patients who would attempt self-harm for the first-time (from VHA CDW and Suicide Prevention Applications Network) were collected as cases. Following previous research that used the ICD-9 codes for intentional self-harm[44,48,49], we converted those ICD-9 into ICD-10CM and defined intentional self-harm using ICD-10CM codes as in Supplementary Data 1. A recent study found that self-harm related ICD-10-CM codes only miss a small proportion of actual self-harm incidents[50]. Another 14 patients were considered as controls who were diagnosed with PTSD but would not attempt self-harm in six months after their last visit. The observed self-harm rate was 1.9% among PTSD patients. Further details of the cohort definition are available in Supplementary Fig. 4. All inpatient and outpatient ICD codes were included in our data.

### Implementation details
For pretraining, we tasked TransformEHR to predict the next visit ICD codes recursively. Specifically, for each patient's input sequence, we used the first visit to predict the second visit, and then used the first two visits to predict the third visit, and so on. This process was repeated until the last visit as recorded in the input sequence was predicted. ICD codes were ordered by their priorities within a visit. We set the maximum sequence length (number of ICD codes) to be 512. To prevent the model from simply memorizing the pretraining data, we randomly drop 15% of visits from each input during pretraining. Other hyperparameters include a warmup ratio of 0.1, a learning rate of 1e − 3, a dropout rate of 0.1, and an L2 weight decay of 1e − 3 with fp16. To make a fair comparison, we pretrained the baseline BERT model on the same pretraining cohort, instead of using existing models pretrained on other cohorts. We used 4 Nvidia Tesla P40 GPU of 22 GB graphics memory capacity and trained each model for 6 days with more than 280 k steps and batch size of 48.

During finetuning, we added a task-specific classifier layer (a linear layer) on top of TransformEHR and BERT to predict disease or outcome as a binary classification task. For a fair comparison with other medical BERT models[13–15], TransformEHR used six layers in both encoder and decoder modules to best match the amount of parameters. All models were finetuned with five random seeds and statistical tests were carried out among these models. To ensure a fair comparison, the same feature transformation, L2 regularization, and hyperparameter tuning strategies, were used to finetune TransformEHR and all baseline models. For each outcome prediction, we built training, validation, and test datasets by the ratio of 7:1:2. One sided student's t test was used to determine if TransformEHR outperforms baseline models. All finetuning experiments were conducted with Nvidia Tesla P40 GPU, and each single disease or outcome prediction was finetuned within 12 h.

### Reporting summary
Further information on research design is available in the Nature Portfolio Reporting Summary linked to this article.

## Data availability

The study protocol was approved by the Institutional Review Board at the VA Bedford Healthcare System under the waiver of informed consent. The study was exempted because the research involves only information collection and analysis involving the investigator's use of identifiable information when that use is regulated under 45 Code of Federal Regulations (CFR) parts 160 and 164, subparts A and E, for the purposes of health care operations or research as those terms are defined at 45 CFR 164.512(b). The VHA EHR data are available under restricted access for Veterans' privacy and data security laws, access can be obtained by relevant approvals through VA Informatics and Computing Infrastructure (VINCI) (contact: VINCI@va.gov). Individuals who wish to use this data for research purposes must fulfill the research credentialing requirements as outlined by the VA Office of Research and Development, with the approval process expected to take from 1 month to 1 year. The MIMIC-IV raw data is publicly available through Physionet. aiming to utilize this data for research will be required to meet research credentialing requirements as outlined at the Physionet's web site: https://physionet.org/content/mimiciv/2.2/. Requests are normally processed within a week. We release a minimum dataset to illustrate the training process on https://github.com/whaleloops/TransformEHR.

## Code availability

Our finetuning code is publicly available on https://github.com/whaleloops/TransformEHR/. Experiments were conducted using Python version 3.8, torch version 1.9.0, transformer library version 4.16.2. Visualization was obtained using Python packages matplotlib version 3.3.2.

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

## Acknowledgements

Support for VA data was provided by the VA Health Services Research and Development Service. H.Y. was supported by R01MH125027, R01DA056470, and R01AG080670, all of which are from the National Institutes of Health (NIH). In addition, she was also supported by I01HX003711 from the US Department of Veterans Affair Veterans Health Administration. Z.Y. was supported by R01MH125027 from NIH. A.M. was supported by R01MH125027 and R01DA056470 from NIH. D.B. was supported by R01AG080670 from NIH. The content is solely the responsibility of the authors and does not necessarily represent the official views of the National Institutes of Health.

## Author contributions

H.Y. initialized the conceptualization of the project. Z.Y. designed the study, implemented the methods, and performed the data analysis. W.L. checked the validity of the data. Z.Y. and A.M. interprete the results with substantial input from D.B. and H.Y. All authors contributed to manuscript preparation.

## Competing interests

The authors declare no competing interests.
