## [Peer Review File · Nature Communications]

TransformEHR: Transformer-based encoder-decoder generative model to enhance prediction of disease outcomes using electronic health recordsREVIEWER COMMENTS

Reviewer #1 (Remarks to the Author):

This interesting work describes training a new clinical foundation model using more than 6 million VA patients' diagnosis data. The model utilized an encoder-decoder transformer architecture and use the next visit codes prediction (DOAP) as the main pretraining task and used the Jaccard similarity score for the pretraining performance evaluation.

However, my first recommendation is to change the model name to a more unique one, as Enformer is the name for Deepemind's gene expression prediction model. Maybe you can add EHR somewhere in the model name.

Second, the use of PHI data like actual visit dates will highly impact both the usability and the generalizability of the model. Instead, I'd highly recommend the authors to use either the day difference between events or if needed the season or the month along with the year

Additionally:

1. It is not clear why the authors only used ICD-10 codes.
2. The AUROC reported only for Pancreatic Cancer in the main manuscript. The supplement tables are showing Jaccard scores, which is weird to use for discriminative accuracy. I'd highly recommend the authors move the main results into the main manuscript, as well as report all AUROC, AUPRC, and other metrics with the best-defined thresholds.
3. I do not see the results in MIMIC IV versus the results on the VA cohorts

Reviewer #2 (Remarks to the Author):

Summary

The authors introduce Enformer, an encoder-decoder model for predicting ICD-10 codes for a future visit based on prior visits. The authors trained and evaluated Enformer using a very large sample of VA EHR data (255M visits by 6.5M patients) and showed that it outperformed multiple baselines for predicting ICD-10 codes, as well as two specific clinically-significant tasks: predicting pancreatic cancer and self-harm in patients with PTSD. Overall, the methods are well-described and appropriate, with ablation analyses to help determine the most useful model components as well as analyzing model performance in different patient subgroups based on demographics. The paper could be strengthened by addressing the comments below.

Major points

1. Line 78: PPV 1.7% really practically useful? If implemented in clinical practice, over 98% of interventions would be for patients who will not actually harm themselves. The statement that PPV 8.45% *minimizes* resource cost and intrusion directed at patients who will never attempt self-harm. (lines 78-79) is certainly not true as stated, though it is certainly an

improvement over the (implausibly low) 1.7% baseline.

2. Please clarify the *meaning* of what you are predicting. ICD-10 (probably more precisely ICD-10-CM) is how the meaning is encoded, but does not fully explain what is being predicted. Are these ICD codes entered for billing purposes (what does that mean in a VA setting)? Clinical purposes (e.g., as problem list entries)? Something else? Without this information, it is hard to understand the contribution of this system beyond numerical comparison to other systems.

3. The VA and MIMIC populations are clinically very different (general vs ICU). It is surprising that a system pretrained on a general population performed well on a very different population. More information regarding the prediction tasks would be helpful. For example, for self-harm in PTSD, were previous ICU admissions used to predict future ICU admissions where self-harm is noted? What about patients who were discharged from ICU, had PTSD and harmed themselves but were not subsequently admitted to the same ICU?

4. The improvements in PPV over BERT are relatively modest in absolute terms. For example, for predicting self-harm with full history: 7.14% \rightarrow 8.67%. Is this clinically significant? E.g., how much does this practically decrease resource utilization in anticipated clinical implementation?

5. The prediction of pancreatic cancer ICD-10 codes one visit prior to the documentation of these codes may not be clinically useful. In other words, the system is predicting whether a human clinician is going to assign a code for pancreatic cancer on the next visit. First, billing codes are known to be poor representations of clinical reality (see comment #2 above). Second, the documentation of a code is not equivalent to the first diagnosis of the disease. Third, (even ignoring the first two points) the duration between visits may differ greatly – predicting that pancreatic cancer will be documented on the next visit in 1 year may be much more useful than if the next visit is in 1 week or 1 month. The relationship between codes and clinical reality depends on the type of visit (inpatient vs. outpatient) as well as the condition. Again, it is important to understand what the ICD-10 codes really represent in the VA system.

Minor points

Line 69-70: "Such pretraining objective can leverage the clinical relations among diseases and outcomes." Is awkward and should be rephrased.

Line 75: "...the improvements for uncommon [*ICD code predictions* or *predictions*] were substantial."

Figure 1: What is "Exam to educational institution"?

Figure 3: "Actual disease *patient* had"

Response to Reviewers

We are grateful for the reviewers' insightful comments and suggestions. We believe these have significantly improved our manuscript. We respond point by point to each comment starting with

→

A short summary of the major changes performed:

- In addition to the original time embedding method, in this revision, we followed the reviewers' recommendations to embed time as the number of days difference between visits (daysdiff) embedding, and updated the results.
- We replaced Jaccard similarity with AUROC as the metric for Disease or Outcome Agnostic Prediction (Pretraining Evaluation)
- We added a new prediction window (pancreatic cancer in 12-36 months after the last visit) for pancreatic cancer onset prediction and updated the results for the new experiment (Results and Methods)
- We clarified our hypotheses and definitions (Introduction and Methods).

Reviewer #1 (Remarks to the Author):

This interesting work describes training a new clinical foundation model using more than 6 million VA patients' diagnosis data. The model utilized an encoder-decoder transformer architecture and use the next visit codes prediction (DOAP) as the main pretraining task and used the Jaccard similarity score for the pretraining performance evaluation.

However, my first recommendation is to change the model name to a more unique one, as Enformer is the name for Deepmind's gene expression prediction model. Maybe you can add EHR somewhere in the model name.

→Thanks for the comment, we now changed the name of our model to TransformEHR.

Second, the use of PHI data like actual visit dates will highly impact both the usability and the generalizability of the model. Instead, I'd highly recommend the authors to use either the day difference between events or if needed the season or the month along with the year
We followed the reviewer's comment by embedding the date using the number of days difference between visits, and named this as daysdiff embedding. We updated the Longitudinal EHRs section accordingly.

Our results did not show significant performance differences using different embedding strategies for time (between date-specific embedding and daysdiff embedding). TransformEHR with daysdiff embedding (AUROC: 84.04 ± 0.10) showed no significant improvement to TransformEHR with date-specific embedding (AUROC: 84.09 ± 0.10). However, we agree that daysdiff embedding provides better protection to patient health information compared to the specific date embedding. Thus, we choose days difference embedding, and use it for later

finetuning experiments. We updated the *Pretraining Evaluation: Disease or Outcome Agnostic Prediction* section accordingly.

Additionally:

1. It is not clear why the authors only used ICD-10 codes.

→We agree with the reviewer that other data (e.g., medications and lab results) would further help improve the performance and we described this in our Limitation section. In our future work, we will include additional variables in our pretrained models to further improve the performance.

In this study, our pretrained models were based on ICD codes due to the following 3 reasons: (1) Computational cost: we only included 27 thousand unique ICD-10-CM codes as our code vocabulary, leaving 10 thousand unique CPT codes and 10 thousand unique drug codes for future work. A big vocabulary size forces the model to have a huge embedding matrix as the input and output layer, which causes a significant increase in both memory and time complexity. Given a quote of \$2.04 per GPU-hour from AWS, the estimated cost of pretraining using only ICD codes is about \$4,700, while the cost of pretraining using all CPT and drug codes is about \$28,600 per run. The pretraining time increases from 6 days to about 38 days for each run in the hyperparameter search (10 runs total). (2) Existing literature: Most published pretrained models were trained on the ICD codes alone [2,3]. (3) Clinical Relevance: Though only pretrained with the ICD codes, our predictive models yielded performance that can be considered as clinically relevant. Specifically, previous research suggests cost-effective health care interventions using ICER thresholds of \$150k per QALY, [1,11] our TransformEHR reduces ICER to \$109k per QALY.

2. The AUROC reported only for Pancreatic Cancer in the main manuscript. The supplement tables are showing Jaccard scores, which is weird to use for discriminative accuracy. I'd highly recommend the authors move the main results into the main manuscript, as well as report all AUROC, AUPRC, and other metrics with the best-defined thresholds.

→We followed the advice and added AUROC in Table 2, 3, and 4. We also removed jaccard scores as metric and to report the AUROC for disease/outcome agnostic prediction tasks. The results are updated in Table 2 in this revision after moving from the supplement as suggested. Visit masking improves prediction by 95%CI: 2.52% - 2.96%, $P < 0.001$ in AUROC. encoder-decoder architecture improves prediction by 95%CI: 0.74% - 1.16%, $P < 0.001$ in AUROC, time embedding improves prediction by 95%CI: 0.01% - 0.43%, $P = 0.021$ in AUROC. Our new results show that the major finding is consistent as the previous version: the contribution from 1) visit masking > 2) encoder-decoder architecture > 3) time embedding.

3. I do not see the results in MIMIC IV versus the results on the VA cohorts

→We include the MIMIC IV results in Supplementary Table 5. Our results show that the TransformEHR model with pretraining improved AUROC (95%CI: 0.8%, 3.6%; $p=0.005$). In comparison, BERT with pretraining outperformed BERT without pretraining in AUROC with limitation (95%CI: -0.3%, 2.7%; $p=0.103$). Our results show that TransformEHR was more generalizable than BERT.

		BERT	TransformEHR
Without pretraining	Mean	77.37	79.36
	Std	±1.07	±0.94
With pretraining	Mean	78.31	81.15
	Std	±0.39	±0.49

Supplementary Table 5. Generalization evaluation of disease/outcome agnostic prediction: AUROC scores on MIMIC-IV dataset when pretrained from VA

Reviewer #2 (Remarks to the Author):

Summary

The authors introduce Enformer, an encoder-decoder model for predicting ICD-10 codes for a future visit based on prior visits. The authors trained and evaluated Enformer using a very large sample of VA EHR data (255M visits by 6.5M patients) and showed that it outperformed multiple baselines for predicting ICD-10 codes, as well as two specific clinically-significant tasks: predicting pancreatic cancer and self-harm in patients with PTSD. Overall, the methods are well-described and appropriate, with ablation analyses to help determine the most useful model components as well as analyzing model performance in different patient subgroups based on demographics. The paper could be strengthened by addressing the comments below.

Major points

1. Line 78: PPV 1.7% really practically useful? If implemented in clinical practice, over 98% of interventions would be for patients who will not actually harm themselves. The statement that PPV 8.45% *minimizes* resource cost and intrusion directed at patients who will never attempt self-harm. (lines 78-79) is certainly not true as stated, though it is certainly an improvement over the (implausibly low) 1.7% baseline.

→We thank the reviewer for this suggestion. In a recent work published by Ross et al [1], they calculated the cost of providing the intervention of cognitive behavioral therapy (CBT) and the potential cost to healthcare sector and society if a self-harm occurs. They concluded that a PPV of 1.7% is a cutoff threshold to consider intervention as cost-effective.

In this revision, we added more details regarding the clinical relevance. Specifically, we rephrased the content starting line 78 to “TransformEHR achieved a positive predictive value (PPV) of 8.8% for prediction of intentional self-harm among the top 10% PTSD patients at high predicted risk. A recent study has shown that a practical suicide prevention tool must achieve above 1.7% PPV to be considered as cost-effective: balance the costs of providing the intervention against the potential health care–related costs if self-harm occurs. A PPV of 8.8% is

substantially above the threshold of 1.7% needed to balance the cost in clinical practice. This shows the potential of deploying TransformEHR for clinical screening and interventions.”

2. Please clarify the *meaning* of what you are predicting. ICD-10 (probably more precisely ICD-10-CM) is how the meaning is encoded, but does not fully explain what is being predicted. Are these ICD codes entered for billing purposes (what does that mean in a VA setting)? Clinical purposes (e.g., as problem list entries)? Something else? Without this information, it is hard to understand the contribution of this system beyond numerical comparison to other systems.

→Our TransformEHR employed a pretrained model based on the prediction of ICD-10-CM codes and was fine-tuned for specific clinical outcomes (e.g., intentional Self-Harm in Patients with PTSD). Although there are data quality issues [4], ICD codes have been widely used to define diseases or outcomes for prediction [5,6,7,8].

In this paper, the ICD codes defined for pancreatic cancer (C25) were taken from previous studies [3,8]. the ICD codes defined for self-harm outcome (in Supplementary Table 6.) were curated from published studies [5,6] (Specifically in *Definition of suicide attempt* section in [5] and *Diagnoses of self-harm* section in [6]). Despite the possible shortcomings and noise that may be inherent in these ICD codes, a recent study found that these ICD-10-CM codes could identify 90% of self-harm events [9]. We modified the *Cohort and Use Case Definitions* section to highlight these sources.

3. The VA and MIMIC populations are clinically very different (general vs ICU). It is surprising that a system pretrained on a general population performed well on a very different population. More information regarding the prediction tasks would be helpful. For example, for self-harm in PTSD, were previous ICU admissions used to predict future ICU admissions where self-harm is noted? What about patients who were discharged from ICU, had PTSD and harmed themselves but were not subsequently admitted to the same ICU?

→To evaluate the generalizability, we used the MIMIC-IV dataset to build a Disease or Outcome Agnostic Prediction (DOAP) dataset, including the prediction of Pneumonia, Heart failure, Gout. Our TransformEHR, which was pretrained using the VA EHR data, performed well on the MIMIC data, even though the two patient populations differed. Our results support that pretrained models are generalizable.

TransformEHR model was designed to learn general patterns from large datasets before being fine-tuned on specific tasks. This pretrain-finetune approach has been shown to work remarkably well in other fields such as natural language processing (NLP), where large language models like ChatGPT and FLAN are pretrained on a diverse range of internet text and then fine-tuned on specific tasks.[10] In the context of our study, the pretrained model was initially trained on a general population (VA). Thus, it learned general patterns of disease progression and healthcare utilization which are not specific to any particular patient group. When the model was subsequently fine-tuned on the MIMIC data, it was able to adapt its

previously learned general patterns to the specific characteristics of the ICU. We referred to this as the pretrain-finetune paradigm in our paper.

4. The improvements in PPV over BERT are relatively modest in absolute terms. For example, for predicting self-harm with full history: 7.14% 8.67%. Is this clinically significant? E.g., how much does this practically decrease resource utilization in anticipated clinical implementation?
→ The improvements in PPV over BERT translates to 12% reduction in resource utilization cost defined in [1]. Specifically, we followed [1] and defined resource utilization cost as incremental cost-effectiveness ratio (ICER) of self-harm risk prediction and intervention: the ratio of its incremental cost to its incremental quality-adjusted life-years (QALYs) compared with usual care. In the US, the previous author defines cost-effective health care interventions using ICER thresholds of \$150k per QALY [1,11]. BERT reduces ICER to \$123k per QALY and TransformEHR reduces ICER to \$109k per QALY. We updated the *discussion* section accordingly.

5. The prediction of pancreatic cancer ICD-10 codes one visit prior to the documentation of these codes may not be clinically useful. In other words, the system is predicting whether a human clinician is going to assign a code for pancreatic cancer on the next visit. First, billing codes are known to be poor representations of clinical reality (see comment #2 above). Second, the documentation of a code is not equivalent to the first diagnosis of the disease. Third, (even ignoring the first two points) the duration between visits may differ greatly – predicting that pancreatic cancer will be documented on the next visit in 1 year may be much more useful than if the next visit is in 1 week or 1 month. The relationship between codes and clinical reality depends on the type of visit (inpatient vs. outpatient) as well as the condition. Again, it is important to understand what the ICD-10 codes really represent in the VA system.

→We used the same setting by Med-BERT for the prediction of pancreatic cancer onset. We agree that such a setting may not be clinically useful. Since the main purpose of our paper is to report an innovative predictive model and demonstrate its performance, it may be best to compare with other state-of-the-art model (Med-BERT) using the same settings.

Regarding the concern about linking ICD codes to disease or outcome, please see the response to comment #2 above.

Following this excellent advice, we have completed additional experiments for the prediction of pancreatic cancer. We set the prediction window to be 12-36 months after the last visit, and set other data filtering criteria the same as Supplementary Figure 2 in Med-BERT[3]. Specifically, we include cases as 4,639 patients of (1) 45 years or older with no report of any other cancer disease before their first pancreatic cancer diagnosis and (2) diagnosis made between 12 and 36 months after their last visit.

We evaluated our TransformEHR and baselines on this dataset and the results are shown in Table 3 below. TransformEHR achieved AUROC of 81.95 (95% CI: 81.06, 82.85) outperforming logistic regression model (73.64; 95% CI: 71.39, 75.90, $p < 0.001$), LSTM (76.98; 95% CI: 76.44, 77.52, $p < 0.001$) and BERT (79.22; 95% CI: 78.75, 79.69, $P < 0.001$). We updated the *Methods*

(Cohort and Use Case Definitions) and Results (Finetuning Evaluation: Single Disease or Outcome Prediction) sections accordingly.

Models		AUROC		AUPRC	
Without Pretraining	Logistic regression	73.64	±2.26	68.95	±2.14
	LSTM	76.98	±0.54	73.48	±0.55
	BERT without pretraining	77.27	±0.45	74.00	±0.31
With Pretraining	BERT	79.22	±0.47	76.89	±0.48
	TransformEHR (ours)	81.95	±0.90	78.64	±0.85

Table 3. Performance of models for pancreatic cancer prediction. Result is calculated from best hyperparameters with 5 randomized seeds each.

Minor points

Line 69-70: “Such pretraining objective can leverage the clinical relations among diseases and outcomes.” Is awkward and should be rephrased.

→Thanks for your comment, this sentence has been rephrased as “Such pretraining objective helps TransformEHR uncover the complex interrelations among different diseases and outcomes.”

Line 75: “...the improvements for uncommon [*ICD code predictions* or *predictions*] were substantial.”

→Keyword added

Figure 1: What is “Exam to educational institution”?

→“Exam to educational institution” is ICD10 code Z02.0.

Figure 3: “Actual disease *patient* had”

→Typo corrected

References:

- [1] Ross et al. Accuracy Requirements for Cost-effective Suicide Risk Prediction Among Primary Care Patients in the US.
- [2] Li et al., BEHRT: Transformer for Electronic Health Records
- [3] Rasmy et al. Med-BERT: pretrained contextualized embeddings on large-scale structured electronic health records for disease prediction.
- [4] Botsis et al. Secondary Use of EHR: Data Quality Issues and Informatics Opportunities
- [5] Zheng et al. Development of an early-warning system for high-risk patients for suicide attempt using deep learning and electronic health records

[6] Simon et al. Predicting Suicide Attempts and Suicide Deaths Following Outpatient Visits Using Electronic Health Records.

[7] Sall et al. Assessment and Management of Patients at Risk for Suicide: Synopsis of the 2019 U.S. Department of Veterans Affairs and U.S. Department of Defense Clinical Practice Guidelines

[8] Placido et al. A deep learning algorithm to predict risk of pancreatic cancer from disease trajectories

[9] Simon et al. Accuracy of ICD-10-CM encounter diagnoses from health records for identifying self-harm events

[10] Chung et al. Scaling Instruction-Finetuned Language Models

[11] Anderson et al. ACC/AHA statement on cost/value methodology in clinical practice guidelines and performance measures: a report of the American College of Cardiology/American Heart Association Task Force on Performance Measures and Task Force on Practice Guidelines.

[12] Nock et al. Prediction of Suicide Attempts Using Clinician Assessment, Patient Self-report, and Electronic Health Records.

REVIEWER COMMENTS

Reviewer #1 (Remarks to the Author):

Thank you for considering our recommendations.

Reviewer #3 (Remarks to the Author):

The authors have conscientiously attempted to address this reviewer's comments, point by point. Regarding the five major points:

1) The clarification (clinical utility vs. cost-effectiveness based on a prior cost-effectiveness analysis) is helpful and addresses the question.

2) Unfortunately, it seems that the authors misunderstood the point. The request for clarification was not related to the coding system (ICD-10 or ICD-10-CM). In other words, the concern was not that the authors attempted to predict ICD-10-CM, but should have predicted something else (e.g., data coded in a different standard such as SNOMED).

Instead, the clarification question related to the meaning of what was being predicted. The authors suggest that by predicting ICD-10-CM codes related to self-harm and pancreatic cancer, the system would (potentially) allow an intervention to prevent the outcome - e.g., Cognitive Behavioral Therapy (CBT) for self-harm (this relates to the 1.7% threshold from the Ross paper cited by the authors). For this to be true, the following must be true:

a) The ICD-10-CM codes must identify self-harm events (the Simon paper cited by the authors provides partial support - note that the 90% is not "accuracy" but the proportion of self-harm codes that are accompanied by evidence of self-harm. As Simon, et al write: "we must acknowledge that some self-harm events would be excluded by our selection of specific codes and we cannot accurately estimate the number of true self-harm events not detected by our methods."

b) The timing of the ICD-10-CM code must coincide with the self-harm event. If these are billing codes, then they may simply reflect visits to psychiatrists due to (for example) concern regarding self-harm (e.g., suicidal ideation) without actual self-harm (e.g., suicide attempt).

These significant limitations should at least be acknowledged.

3) Again, I am not sure that the response addresses the reviewer's question. The authors response seems to address model generalizability from a purely data perspective. I.e., given data set #1 and data set #2, with inputs and outcomes in each data set, does the model trained on data set #1 perform the same with respect to data set #2. This was not the question.

Instead, the question (as in #2) related to the meaning of ICD-10-CM codes. In many (but not all) cases, ICD-10-CM codes are assigned for billing purposes. However, VA providers do not bill in the same way as providers outside of the VA system. Thus, the question related to:

a) Why the ICD-10-CM codes were assigned by the VA. Was it for billing purposes? Was it for some other purposes (see: <https://www.ncbi.nlm.nih.gov/pmc/articles/PMC4558479/#:~:text=ICD codes are integral to,records by disease and operations%2C>) – which codes were used? Who assigned them? Etc. Some clarification beyond the “we have these data” would be helpful.

b) MIMIC is an ICU data set. Again, it would be helpful to understand what ICD codes were used and it seems that the kinds of codes assigned to an ICU population would be different from those in a general population. For example, patients are not really expected to return to the ICU in the same way that they would be expected to follow up with their outpatient (e.g., primary care) physician. Thus, predicting codes for future ICU admissions seems like an odd choice of prediction tasks.

4) The additional estimate of (improved) cost-effectiveness is helpful and addresses the question.

5) The additional analyses are helpful and address the question.

Response to Reviewers

We are grateful for the comments and suggestions from reviewers and editors. We colored our point-by-point responses for better context. The reviewer question is in black, our response to the first round of review (we label as “initial response” and “initial question”) is in blue, and our response to this round of review (we label as “new response” and “followup question”) is in red.

Reviewer #1 (Remarks to the Author):

Reviewer #2 (Remarks to the Author):

Initial question #2 from reviewer.

“Please clarify the *meaning* of what you are predicting. ICD-10 (probably more precisely ICD-10-CM) is how the meaning is encoded, but does not fully explain what is being predicted. Are these ICD codes entered for billing purposes (what does that mean in a VA setting)? Clinical purposes (e.g., as problem list entries)? Something else? Without this information, it is hard to understand the contribution of this system beyond numerical comparison to other systems.”

Initial response: Our TransformEHR employs a pretrained model based on the prediction of ICD-10-CM codes and is fine-tuned for specific clinical outcomes (e.g., intentional Self-Harm in Patients with PTSD). Although there are data quality issues , **ICD codes have been widely used to define diseases or outcomes to be predicted.**

In this paper, ICD codes defined for pancreatic cancer (C25) were taken from previous studies. the ICD codes defined for self-harm outcome (in Supplementary Table 6.) were curated from published studies. Despite the possible shortcomings and noise that may be inherent in these ICD codes, **a recent study found that these ICD-10-CM codes could identify 90% of self-harm events.** All ICD-10-CM codes extracted for this study are for clinical purposes, specifically, as problem list entries. We modified the *Cohort and Use Case Definitions* section to highlight these sources.

Followup question: “Unfortunately, it seems that the authors misunderstood the point. The request for clarification was not related to the coding system (ICD-10 or ICD-10-CM). In other words, the concern was not that the authors attempted to predict ICD-10-CM, but should have predicted something else (e.g., data coded in a different standard such as SNOMED).”

New response: We agree that SNOMED is a much better clinical representation than the ICD codes (ICD-10-CM disease codes in this work). However, SNOMED codes are not widely assigned to EHRs, but ICD codes are. Therefore we used ICD codes.

Followup question: “Instead, the clarification question related to the meaning of what was being predicted. The authors suggest that by predicting ICD-10-CM codes related to self-harm and pancreatic cancer, the system would (potentially) allow an intervention to prevent the outcome - e.g., Cognitive Behavioral Therapy (CBT) for self-harm (this relates to the 1.7% threshold from the Ross paper cited by the authors). For this to be true, the following must be true:”

“a) The ICD-10-CM codes must identify self-harm events (the Simon paper cited by the authors provides partial support - note that the 90% is not "accuracy" but the proportion of self-harm codes that are accompanied by evidence of self-harm. As Simon, et al write: ‘we must acknowledge that some self-harm events would be excluded by our selection of specific codes and we cannot accurately estimate the number of true self-harm events not detected by our methods.’”

“b) The timing of the ICD-10-CM code must coincide with the self-harm event. If these are billing codes, then they may simply reflect visits to psychiatrists due to (for example) concern regarding self-harm (e.g., suicidal ideation) without actual self-harm (e.g., suicide attempt).”

New response: We agree with the reviewer and acknowledge the limitation of ICD codes for self-harm in the *Limitation* section, as shown below. Regarding suicidal ideation, as shown in Supplement Table 6, we followed the guideline [1] to include only T14.91XA and excluding R45.851 (suicidal ideation) for patients with first-time intentional self-harm.

We added the following to the *Limitation* section:

While we followed the previous studies to use ICD-10-CM codes to identify intentional self-harm,[2,3] we acknowledge that the ICD-10-CM representation of self-harm would miss some patients who conducted self-harm. [4] In addition, the date of the encounter may not be the actual date of the self-harm.

Initial question #3 from the reviewer:

“The VA and MIMIC populations are clinically very different (general vs ICU). It is surprising that a system pretrained on a general population performed well on a very different population. More information regarding the prediction tasks would be helpful. For example, for self-harm in PTSD, were previous ICU admissions used to predict future ICU admissions where self-harm is noted? What about patients who were discharged from ICU, had PTSD and harmed themselves but were not subsequently admitted to the same ICU?”

Initial response: To test the generalizability, we used the MIMIC-IV dataset to build a non-VA Disease or Outcome Agnostic Prediction (DOAP) dataset. Our TransformEHR, which was pretrained using the VA EHR data, performed well on the MIMIC data, even though the two patient populations differ. Our results **support the generalizability of incorporating pretrained models**.

TransformEHR model was designed to learn general patterns from large datasets before being fine-tuned on specific tasks. This approach has been shown to work remarkably well in other fields such as natural language processing (NLP), where large language models like ChatGPT and FLAN are pretrained on a diverse range of internet text and then fine-tuned on specific tasks.[10] In the context of our study, **the pretrained model was initially trained on a general population (VA)**. Thus, it learned general patterns of disease progression and healthcare utilization which are not specific to any particular patient group. When the model was **subsequently fine-tuned on the MIMIC data**, it was able to adapt its previously learned general patterns to the specific characteristics of the ICU. We referred to this as the **pretrain-finetune paradigm** in our paper.

Followup question: Again, I am not sure that the response addresses the reviewer’s question. The authors response seems to address model generalizability from a purely data perspective. I.e., given data set #1 and data set #2, with inputs and outcomes in each data set, does the model trained on data set #1 perform the same with respect to data set #2. This was not the question.

Instead, the question (as in #2) related to the meaning of ICD-10-CM codes. In many (but not all) cases, ICD-10-CM codes are assigned for billing purposes. However, VA providers do not bill in the same way as providers outside of the VA system. Thus, the question related to:

a) Why the ICD-10-CM codes were assigned by the VA. Was it for billing purposes? Was it for some other purposes (see:

<https://www.ncbi.nlm.nih.gov/pmc/articles/PMC4558479/#:~:text=ICD codes are integral to,records by disease and operations%2C>) – which codes were used? Who assigned them? Etc. Some clarification beyond the “we have these data” would be helpful.

New response: The ICD codes used in the VA were assigned by VA medical record technician [5]. The ICD codes serve several important purposes including clinical studies, performance measurement, workload capture and operation, cost determination, and billing. [6] We added this statement in the *data* section.

b) MIMIC is an ICU data set. Again, it would be helpful to understand what ICD codes were used and it seems that the kinds of codes assigned to an ICU population would be different from those in a general population. For example, patients are not really expected to return to the ICU in the same way that they would be expected to follow up with their outpatient (e.g., primary care) physician. Thus, predicting codes for future ICU admissions seems like an odd choice of prediction tasks.

New response: In our work, we used the complete ICD codes assigned to a patient visit in both MIMIC and VA EHR data. We did not exclude any ICD codes. Although we acknowledge that the VA has both ICU (VA Medical Intensive Unit) and emergency department (e.g., West Roxbury VA Medical Center), even though a majority of VA facilities are in primary care, we agree that the distribution of the ICD codes assigned to an ICU population (MIMIC) would be different from the ICD distribution in the VA, as shown in the Venn diagram. And added this to the *Generalizability Evaluation* section.

We also agree with the reviewer that patient follow up in an ICU setting is very different from patient follow up in an outpatient setting. We chose the MIMIC EHR dataset in part because it is currently the only publicly available EHR dataset. Despite the substantial differences between MIMIC and VA clinical settings, our model, TransformEHR which was trained in the VA EHR dataset, has shown to consistently outperform the state-of-the-art BERT model, on the MIMIC EHR dataset. Our TransformEHR models also showed consistently to be superior to the BERT models when trained and validated using different clinics at the VA. We may conclude that TransformEHR outperforms BERT regardless of the similarity of patient distribution between the clinical setting where the models are trained and the setting where the models are deployed.

Supplementary Figure 4. Venn diagram on the unique ICD-10-CM codes in the MIMIC-IV data and our VA data. Common codes in MIMIC-IV include: Hypertension(I10), Hyperlipidemia(E785), Gastroesophageal reflux disease (K219), Anxiety (F419), Arteriosclerosis (I2510), Acute kidney failure (N179), Type 2 diabetes mellitus without complications (E119), Hypothyroidism (E039), Obstructive Sleep Apnea (G4733), Unspecified atrial fibrillation (E4891).

References:

- [1] ICD-10-CM Coding for Suicide Attempts and Suicidal Ideation. URL: https://www.docs.lms.va.gov/LMSDocs/Docs/117288/ICD-10-CMcodingtraining_revisedPP8-31-172.pdf
- [2] Rawat et al. Intentional Self-Harm Among US Veterans With Traumatic Brain Injury or Posttraumatic Stress Disorder: Retrospective Cohort Study From 2008 to 2017
- [3] Patrick et al. Identification of Hospitalizations for Intentional Self-Harm when E-Codes are Incompletely Recorded.
- [4] Simon et al. Accuracy of ICD-10-CM encounter diagnoses from health records for identifying self-harm events.
- [5] Medical records technician - VA handbook 5005/122. URL: https://www.va.gov/vapubs/viewPublication.asp?Pub_ID=1116&FType=2
- [6] Weems et al. Results from the Veterans Health Administration ICD-10-CM/PCS Coding Pilot Study